

# A novel attention-based deep learning model for improving sentiment classification after the case of the 2023 Kahramanmaras/Turkey earthquake on Twitter

Serpil Aslan[1] and Muhammed Yildirim[2]

[1] Software Engineering, Malatya Turgut Ozal University, Malatya, Turkey
[2] Computer Engineering, Malatya Turgut Ozal University, Malatya, Turkey

## ABSTRACT

Twitter has emerged as one of the most widely used platforms for sharing information and updates. As users freely express their thoughts and emotions, a vast amount of data is generated, particularly in the aftermath of disasters, which can be collected quickly and directly from individuals. Traditionally, earthquake impact assessments have been conducted through field studies by non-governmental organizations (NGOs), a process that is often time-consuming and costly. Sentiment analysis (SA) on Twitter presents a valuable research area, enabling the extraction and interpretation of real-time public perceptions. In recent years, attention-based methods in deep learning networks have gained significant attention among researchers. This study proposes a novel sentiment classification model, MConv-BiLSTM-GAM, which leverages an attention mechanism to analyze public sentiment following the 7.8 and 7.5 Mw earthquakes that struck Kahramanmaraş, Turkey. The model employs the FastText word embedding technique to convert tweets into vector representations. These vectorized inputs are then processed by a hybrid model integrating convolutional neural networks (CNNs) and recurrent neural networks (RNNs) with a global attention mechanism. This ensures careful consideration of semantic dependencies in sentiment classification. The proposed model operates in three stages: (i) MConv—Local Contextual Feature Extraction, (ii) bidirectional long short-term memory (BiLSTM)—sequence learning, and (iii) Global Attention Mechanism (GAM)—Attention Mechanism. Experimental results demonstrate that the model achieves an accuracy of 93.32%, surpassing traditional deep learning models in the literature by approximately 3%. This research aims to provide objective insights to policymakers and decision-makers, facilitating adequate support for individuals and communities affected by disasters. Moreover, analyzing public sentiment during earthquakes contributes to understanding societal responses and emotional trends in disaster scenarios.

Corresponding author
Muhammed Yildirim,
muhammed.yildirim@ozal.edu.tr

# INTRODUCTION

With the development of online technologies, a large amount of data is being produced by Internet users, and this increases the demand for data processing and computer-aided analysis (*Wang et al., 2020*; *Shi et al., 2011*). Thanks to the widespread use of the internet, millions of users have access to the data produced every day, but converting this raw data into meaningful information requires various analysis techniques (*Aslan, 2023*). Social media data analysis is important in many areas, from marketing strategies to crisis management. It is possible to determine trends and analyze user behavior by processing text, images, and videos shared on platforms. While increasing social media use allows individuals to share content that can impact society, the healthy analysis of this data depends on using correct data collection methods and reliable analysis tools.

Twitter is one of the most popular social media platforms, and millions of users worldwide express their feelings, thoughts, and reactions (*Pak & Paroubek, 2010*). The presentation of shared large-scale data in a compact and casual language makes Twitter a valuable resource for sentiment analysis (SA) in times of crisis. While artificial intelligence significantly contributes to solving human and societal problems, natural language processing (NLP) also helps computers understand human-generated texts (*Acheampong, Wenyu & Nunoo-Mensah, 2020*). Examining data shared on platforms such as Twitter with NLP-based SA provides important insights into evaluating public reactions, determining individual attitudes, and detecting negative situations such as online bullying (*Tam, Said & Tanriöver, 2021*; *Bashir et al., 2021*). SA, especially during natural disasters, contributes to understanding the psychological states of affected individuals and helps crisis management and intervention processes to be carried out more effectively. In addition, analyses performed on Twitter data are considered an effective method for determining and predicting public opinion trends.

Natural disasters, particularly earthquakes, have severe impacts on individuals and societies, making effective crisis response and impact mitigation essential. Understanding public reactions, identifying needs, and supporting decision-making processes are critical for post-disaster management (*Contreras et al., 2022*). While qualitative methods such as interviews, focus groups, and participatory mapping are commonly used to assess non-physical recovery aspects (*Schumann, 2018*), these approaches are often costly and limited in scope due to their dependence on fieldwork (*Contreras, Wilkinson & James, 2021*). With the rise of social media, information flow and disaster response have significantly evolved. Platforms like Twitter now serve as vital tools for large-scale data collection and damage assessment, as seen in the 2020 Zagreb and Aegean earthquakes (*Contreras et al., 2021*; *Aktas et al., 2021*), enhancing public participation and access to information in emergency management (*Kropivnitskaya et al., 2017*, *2018*; *Simon, Goldberg & Adini, 2015*). Analysis of post-earthquake social media data contributes to faster and more effective coordination of relief and rescue efforts (*Crooks et al., 2013*), while also allowing for the understanding of public reactions through emotional content analysis (*Wu & Cui, 2018*). However, despite the 7.8 and 7.5 Mw earthquakes in and around Kahramanmaraş, Turkey, on February 6, 2023 (*Avcil et al., 2024*) being described

worldwide as the "Disaster of the Century," research on public opinions is limited. Thousands of tweets shared on Twitter immediately after the earthquake provided information on the severity of the earthquake, the affected areas, and urgent needs. This platform played a critical role in finding disaster victims, transmitting calls for help, and coordinating rescue teams; it also allowed individuals to report that they were safe and to submit requests for help. Thus, Twitter contributed to the organization of post-disaster relief processes and was an important communication tool that strengthened social solidarity.

It is essential to comprehend the overall state of affairs following the earthquake, mainly how society reacts to tragedies, fulfills their requests, expedites decision-making, and offers earthquake victims both material and moral support (*Contreras et al., 2022*; *Tehseen, Farooq & Abid, 2020*). This study presents a novel sentiment classification model (MConv-BiLSTM-GAM) to analyze the societal emotional impact of the two major earthquakes that struck Kahramanmaraş, Turkey.

## Motivation and contributions

The motivation and contributions of the proposed study are given below:

- **Original dataset and advanced preprocessing:** In this study, a large dataset of tweets shared in the immediate aftermath of the February 6, 2023, Kahramanmaraş/Turkey earthquake was collected from Twitter. The raw data underwent advanced preprocessing to remove noise, apply lemmatization, and extract meaningful information. Deep learning-based NLP techniques were then used to analyze the earthquake's psychological impact on a global scale.

- **A novel hybrid model (MConv-BiLSTM-GAM):** The proposed model combines convolutional neural networks (CNN), bidirectional long short-term memory (BiLSTM), and the Global Attention Mechanism (GAM) to harness their complementary capabilities for enhanced sentiment classification. FastText embeddings are utilized to convert tweets into vector representations, offering a semantic advantage by operating at the character n-gram level. This allows for better handling of misspellings, morphologically rich forms, and out-of-vocabulary (OOV) words—common features in noisy social media content. The MConv stage applies multiple 1D convolutional filters (kernel sizes 2 and 3) to extract local contextual features, while the BiLSTM layer captures temporal and bidirectional dependencies. To address potential limitations in focusing on key semantic elements, the GAM component assigns attention weights across the sequence, enhancing the model's ability to emphasize relevant tokens and manage long-distance dependencies. This integrated design ensures a robust and context-aware sentiment classification approach, well-suited for informal and unstructured Twitter data.

- **Contribution to post-disaster psychological impact analysis:** This study presents an NLP-based framework for analyzing societal emotions following a disaster, utilizing social media data to assess large-scale sentiment trends. By capturing and interpreting public reactions, the proposed model provides valuable insights for crisis management

and psychological support planning. Its data-driven methodology enhances decision-making processes, facilitating the development of targeted interventions to support affected communities more effectively.

The following sections in the article are organized as follows: "Related Works" presents a literature review of relevant work in the field. "The Proposed System" describes the dataset collection process, the applied preprocessing steps, and the architecture of the proposed model. "Discussions and Experimental Results" focuses on sentiment visualization, data analysis, and a comparative evaluation of the experimental results. Finally, the study is concluded in "Conclusions".

## RELATED WORKS

NLP, a core field of artificial intelligence, enables interaction between humans and machines through the processing of textual or verbal data. With the exponential growth of online communication, effective analysis of large-scale text has become essential (*Shi et al., 2011*). Sentiment analysis (SA), a subfield of NLP, focuses on detecting individuals' attitudes and emotions toward events or entities (*Medhat, Hassan & Korashy, 2014*), with applications in areas such as product evaluation, market research, and audience sentiment assessment (*Li, Goh & Jin, 2020*; *Zheng, Wang & Gao, 2018*). SA techniques are generally categorized into deep learning-based, dictionary-based, and machine learning-based approaches (*Aslan, 2022*). Among these, deep learning methods have demonstrated superior performance due to their ability to automatically extract complex and relevant features (*Aslan, Kızıloluk & Sert, 2023*).

SA, feature selection is vital for accurately detecting emotional expressions, directly influencing model accuracy and generalization (*Zheng, Wang & Gao, 2018*; *Zhang, Wang & Liu, 2018*). Traditional approaches, such as *Zheng, Wang & Gao*'s *(2018)* use of term frequency–inverse document frequency (TF-IDF) with SVM and n-gram-based weighting, have shown limitations due to dependence on manually defined features. Similarly, *Go, Bhayani & Huang (2009)* applied naïve Bayes, MaxEnt, and SVM with various n-gram models, reporting superior results with SVM. To overcome these limitations, recent studies have focused on deep learning-based models integrating word embeddings. *Abid et al. (2019)* proposed a GloVe-Bi-GRU-CNN model, while *Yoon & Kim (2017)* introduced a CNN-BiLSTM using Word2Vec. *Kamyab, Liu & Adjeisah (2021)* combined GloVe embeddings with TF-IDF for enhanced feature representation. Although word embeddings improve semantic understanding, many approaches still underrepresent emotional cues (*Araque et al., 2017*), highlighting the need for methods that balance semantic and affective information in deep learning-based SA (*Kamyab, Liu & Adjeisah, 2021*).

Attention mechanisms are essential components in deep learning, allowing models to selectively focus on the most relevant parts of input data rather than processing all information equally (*Vaswani et al., 2017*). When integrated with RNNs, attention mechanisms enhance performance across various applications (*Bahdanau, Cho & Bengio, 2014*) and are typically categorized as global, self, or hierarchical attention.

*Basiri et al. (2021)* introduced ABCDM, an attention-based model combining Bi-GRU and BiLSTM to capture bidirectional temporal dependencies. *Wen & Li (2018)* proposed ARC, a hybrid RNN-CNN-attention model for sentiment analysis of tweets, effectively capturing both sequential and global features. *Yang et al. (2016)* developed a hierarchical attention network (HAN) that applies attention at both sentence and word levels, while *Liu & Guo (2019)* utilized attention in a BiLSTM-CNN model to emphasize critical hidden layer information. Similarly, *Ma et al. (2017)* presented an interactive attention network to enhance contextual representation. Collectively, these studies demonstrate that attention-based models significantly improve the ability to extract and prioritize relevant semantic information.

Twitter-based sentiment analysis (SA) in disaster contexts serves as a valuable tool for understanding public emotions and improving crisis response. *Ruz, Henríquez & Mascareño (2020)* utilized Bayesian networks on Spanish-language datasets from the 2010 Chile earthquake and the 2017 Catalan referendum, achieving competitive results compared to SVM and random forests. *Behl et al. (2021)* trained supervised models on data from the Nepal and Italy earthquakes, reaching 83.0% accuracy on COVID-19 test data using an optimized MLP. *Mendon et al. (2021)* applied a hybrid framework combining ML and lexicon-based methods, obtaining 81.84% performance on 243,746 Kerala-related tweets. *Yao & Wang (2020)* proposed DSSA-H, combining RF and DANN classifiers for hurricane-related tweets, achieving up to 82.61% accuracy. *Song & Huang (2021)* introduced SentiBERT-BiLSTM-CNN, reaching a 92.75% F1-score. Despite promising outcomes, a gap remains in deep learning-based SA applied from the onset of disasters. Addressing this, *Anthony, Hoi Ki Wong & Joyce (2024)* used FastText-based long short-term memory (LSTM) and CNN models to analyze post-earthquake sentiment, achieving up to 86% accuracy. *Henríquez (2024)* reported 84.29% accuracy using the edRVFL model on Chilean and Catalan tweets. *Alharm & Naim (2024)* employed a BERT-LSTM model for the 2023 Turkey earthquake, attaining 85.43% accuracy. *Blomeier, Schmidt & Resch (2024)* fine-tuned BERT for classifying semantic relevance in German flood-related tweets, achieving 71.0% accuracy. These studies collectively highlight the growing relevance and evolving sophistication of SA techniques in disaster management.

## THE PROPOSED SYSTEM

This work proposes a multilayer deep learning model based on a global attention mechanism to investigate the attention mechanism's capability for sentiment classification following FastText word embedding. Figure 1 displays the general flow diagram of the proposed architecture.

### Dataset and preprocessing

This study utilizes Twitter (*Alam et al., 2021*) as a data source to examine the societal impact of the February 6, 2023 Kahramanmaraş/Turkey earthquake. A total of 215,446 publicly available English tweets were collected *via* MAXQDA software (*MAXQDA, 2020*), covering the period from February 6 to April 27, 2023. To ensure ethical compliance, no personal or identifiable information was included. Relevant tweets were retrieved using

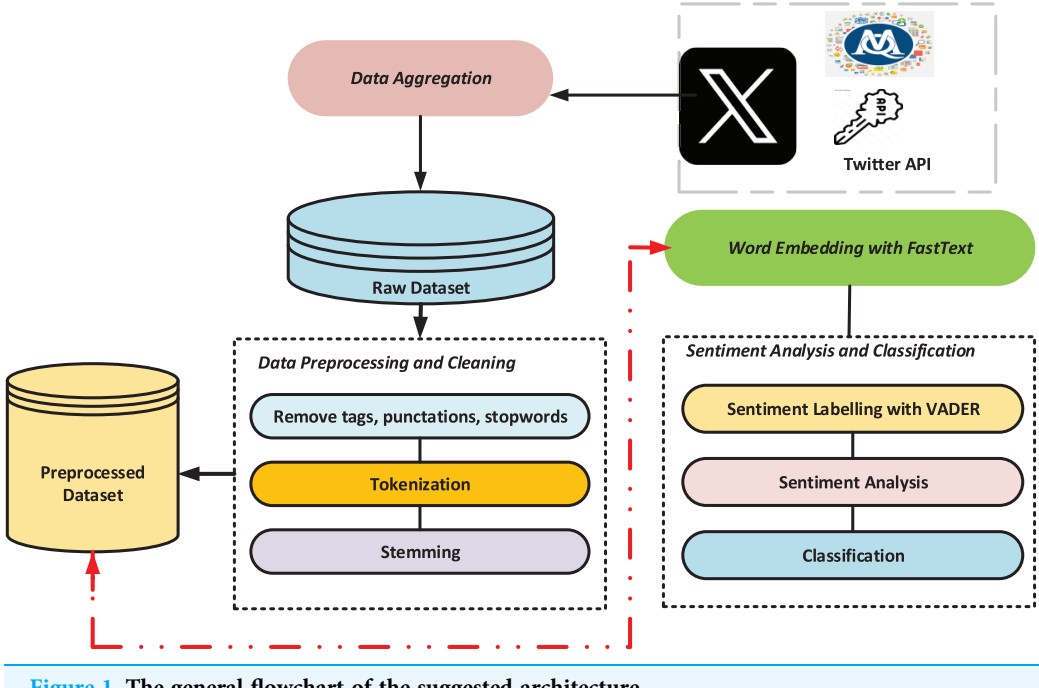

**Figure 1** The general flowchart of the suggested architecture.

common earthquake-related hashtags such as #earthquake, #turkeyearthquake, and #turkeysyriaearthquake. After removing redundant elements (*e.g.*, special characters, links, emojis), preprocessing reduced the dataset to 81,797 clean tweets suitable for sentiment analysis. VADER was used to determine sentiment polarities: 38.91% were negative (31,824 tweets), 37.09% positive (30,336), and 24.01% neutral (19,637). This preprocessing step enhanced both the accuracy and efficiency of the sentiment classification task.

## The proposed model

This study proposes a novel sentiment classification approach based on the MConv-BiLSTM-GAM model to improve performance on Twitter data. As illustrated in Fig. 2, the architecture integrates FastText embeddings with a hybrid structure combining CNN, BiLSTM, and GAM. The model consists of three main stages: MConv for local feature extraction using multiple Conv1D layers with kernel sizes 2 and 3, followed by MaxPooling, BiLSTM for capturing sequential dependencies, and GAM for focusing on semantically relevant information across the sequence. GAM enables the model to attend to distant yet important tokens by learning contextual relevance. Finally, the attention-refined output is passed through two dense layers, concluding with a softmax-based classification layer.

The proposed MConv-BiLSTM-GAM model architecture was optimized using a set of carefully selected hyperparameters for each stage. In the feature extraction stage (MConv), two 1D convolutional layers with 100 filters and kernel sizes of 2 and 3 were employed to capture diverse n-gram features, followed by ReLU activation, a max-pooling layer with a size of 2, and a dropout rate of 0.25 to prevent overfitting. In the sequence learning stage

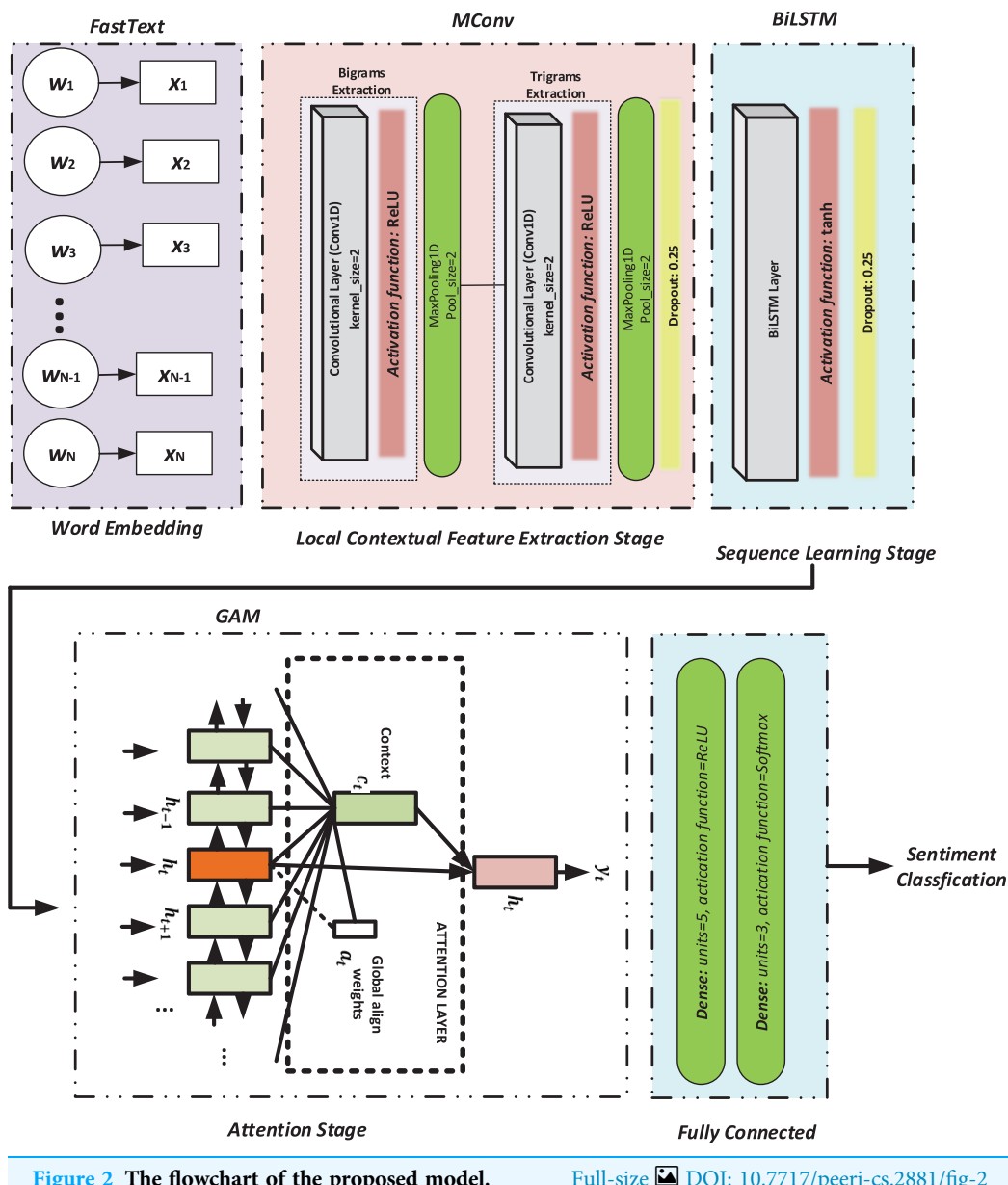

**Figure 2 The flowchart of the proposed model.**

(BiLSTM), a bidirectional LSTM layer with 128 nodes and ReLU activation was used, and a dropout rate of 0.25 was used to maintain regularization. The attention mechanism (GAM) was configured with an input size of 95 to compute attention weights across the sequence. Finally, the classification stage consisted of two fully connected dense layers: the first with five units and ReLU activation, and the second with three units and a softmax activation function to output the sentiment class probabilities. These hyperparameter choices were determined through empirical tuning to effectively balance model complexity and performance.

# DISCUSSIONS AND EXPERIMENTAL RESULTS

The Kahramanmaraş/Turkey earthquake dataset from February 6, 2023, was used in this section's practical experiments to evaluate the proposed model's functionality, assess its robustness, and compare its performance with other deep learning techniques to improve SA accuracy. The experiments were conducted using the Python programming language on the Google Collaborate Pro platform. The study utilized Python libraries, including Pandas, Keras, Numpy, spaCy, and Sklearn. All experiments were tested on a computer with an Intel Core i7 processor, Windows 10 operating system, and 16 GB RAM. The results obtained were examined in this section.

## Analyzing the sentiment distributions

To explore the psychological context of emotional tendencies, word cloud visualization was used to highlight the most frequent and sentiment-relevant terms in tweets. Figure 3 displays the word clouds corresponding to positive, negative, and neutral tweet categories. As shown in Fig. 3, non-emotive but high-frequency words such as "turkey," "earthquake," and "people" were excluded to better emphasize emotionally significant keywords. The word clouds for positive, negative, and neutral sentiment categories reveal distinct linguistic patterns. Despite similar proportions, positive tweets predominantly include expressions of support, solidarity, and reassurance, while negative tweets contain fear-inducing and destructive language. Such negative discourse during disasters may contribute to long-term psychological effects on affected communities. The findings illustrated in Fig. 3 offer valuable insights for decision-makers to design timely and targeted psychological, social, and financial interventions based on the collective emotional state.

## Performance evaluation of the sentiment classification model

This section presents a comparative performance analysis of sentiment classification models on Twitter data related to the February 6, 2023 Kahramanmaraş earthquake. The proposed MConv-BiLSTM-GAM model is evaluated against three baseline deep learning architectures, using both FastText and GloVe embeddings to assess embedding effectiveness. Performance is measured through accuracy, precision, recall, and F1-score (_Aslan & Kaya, 2018_), while overfitting risk is monitored _via_ training and testing accuracy-loss curves (_Reagen et al., 2018_). Additionally, a confusion matrix is used to examine misclassifications. Results demonstrate the proposed model's superior performance, particularly in handling emotionally charged disaster-related content.

The performance of the proposed MConv-BiLSTM-GAM model was thoroughly evaluated using earthquake-related Twitter data, with detailed results provided in Table 1. When utilizing FastText embeddings, the model achieved an average accuracy of 93.32%, outperforming other deep learning models by approximately 3%. The F1-scores for positive, negative, and neutral sentiment classes were 93.48%, 93.46%, and 92.86%, respectively, demonstrating both high accuracy and classification consistency. Comparative analyses show that the CNN model performed relatively poorly, especially in

**Common Words Among Most Positive Tweets**

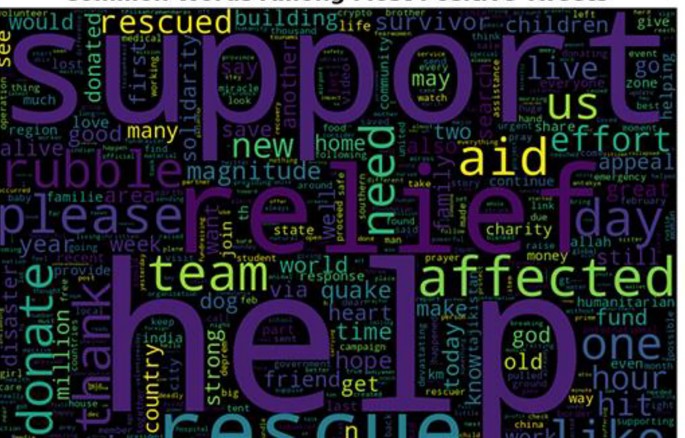

**Common Words Among Most Negative Tweets**

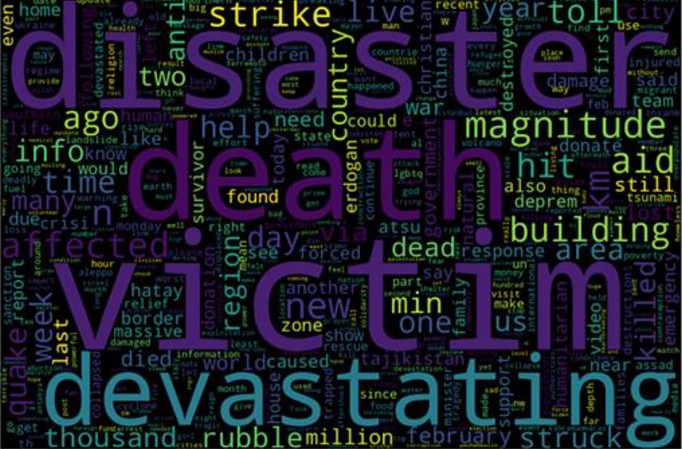

**Common Words Among Most Neutral Tweets**

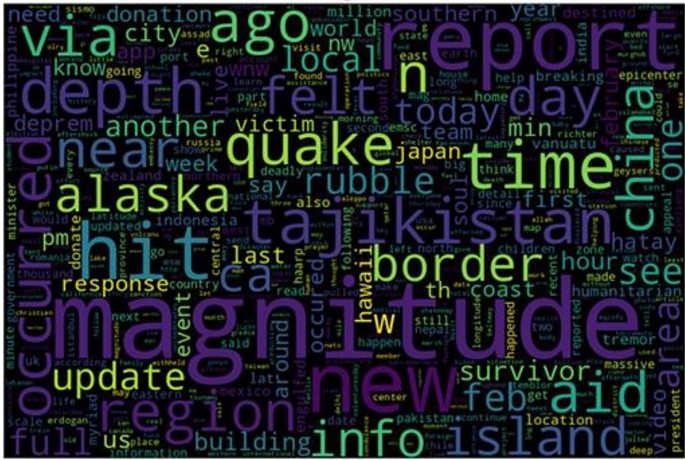

**Figure 3** The word cloud representation of the positive, negative and neutral tweets.

the negative sentiment class, with a precision of 89.72% and a recall of 92.84%. While LSTM and BiLSTM models performed better due to their ability to capture sequential dependencies, the proposed MConv-BiLSTM-GAM model achieved the highest performance. This improvement stems from the synergy between CNN's local feature extraction, BiLSTM's contextual learning, and the Global Attention Mechanism's ability to focus on semantically relevant information. In particular, the MConv layer enhances local pattern recognition through multiple kernel sizes, while BiLSTM captures bidirectional dependencies, and GAM further refines feature relevance, leading to more accurate sentiment classification.

Given that social media data shared during disasters such as earthquakes contain high emotional intensity and similar word patterns, using only CNN or BiLSTM is insufficient for precise SA. Therefore, the extracted features are processed through the GAM mechanism to prioritize the most meaningful information. Experimental results confirm the proposed model's superior performance in sentiment classification.

**Table 1 Comparison of the proposed model's performance with baseline deep learning models.**

| Embedding model | Classification model | Normalized classes 0 → Negative 1 → Neutral 2 → Positive | Performance evaluation metrics | | | Average value |
|---|---|---|---|---|---|---|
| | | | Precision (%) | Recall (%) | F1-Score (%) | Accuracy (%) |
| FastText | MConv-BiLSTM-GAM | 0 | 92.75 | 94.18 | 93.46 | 93.32 |
| | | 1 | 93.53 | 92.20 | 92.86 | |
| | | 2 | 93.82 | 93.14 | 93.48 | |
| | CNN | 0 | 89.72 | 92.84 | 91.25 | 90.55 |
| | | 1 | 88.56 | 92.15 | 90.32 | |
| | | 2 | 92.93 | 87.12 | 89.93 | |
| | LSTM | 0 | 89.56 | 93.12 | 91.30 | 90.92 |
| | | 1 | 91.83 | 89.39 | 90.59 | |
| | | 2 | 91.88 | 89.59 | 90.72 | |
| | BiLSTM | 0 | 91.31 | 91.96 | 91.64 | 91.12 |
| | | 1 | 90.19 | 91.19 | 90.69 | |
| | | 2 | 91.54 | 90.20 | 90.86 | |
| GloVe | MConv-BiLSTM-GAM | 0 | 89.45 | 90.90 | 90.17 | 89.26 |
| | | 1 | 88.48 | 88.07 | 88.27 | |
| | | 2 | 89.57 | 88.30 | 88.93 | |
| | CNN | 0 | 87.06 | 88.31 | 87.68 | 85.67 |
| | | 1 | 78.51 | 88.79 | 83.34 | |
| | | 2 | 89.76 | 80.92 | 85.11 | |
| | LSTM | 0 | 89.29 | 88.88 | 89.08 | 87.96 |
| | | 1 | 86.72 | 86.31 | 86.51 | |
| | | 2 | 87.37 | 88.06 | 87.71 | |
| | BiLSTM | 0 | 87.54 | 91.23 | 89.35 | 88.06 |
| | | 1 | 88.64 | 84.43 | 86.48 | |
| | | 2 | 88.29 | 87.04 | 87.66 | |

Another important factor in the success of the proposed model is the use of the FastText word embedding method. To evaluate its effectiveness, the MConv-BiLSTM-GAM model was tested using datasets vectorized with both FastText and GloVe embeddings, with the results illustrated in Fig. 4. As shown, FastText led to higher performance, with validation accuracy starting at 90% and reaching 93.43% by the end of training, compared to 84% to 87.08% with GloVe. The average accuracy achieved using FastText (93.32%) is approximately 6.35% higher than that of GloVe (87.08%), indicating a notable performance advantage. FastText's ability to capture subword and character-level information through character n-grams makes it particularly effective for handling the morphologically rich and noisy language often found in social media posts during disasters. This allows for more accurate representation of misspelled or informal words. Furthermore, analysis of the accuracy-loss curves suggests that training progresses smoothly without signs of overfitting. This is supported by the application of

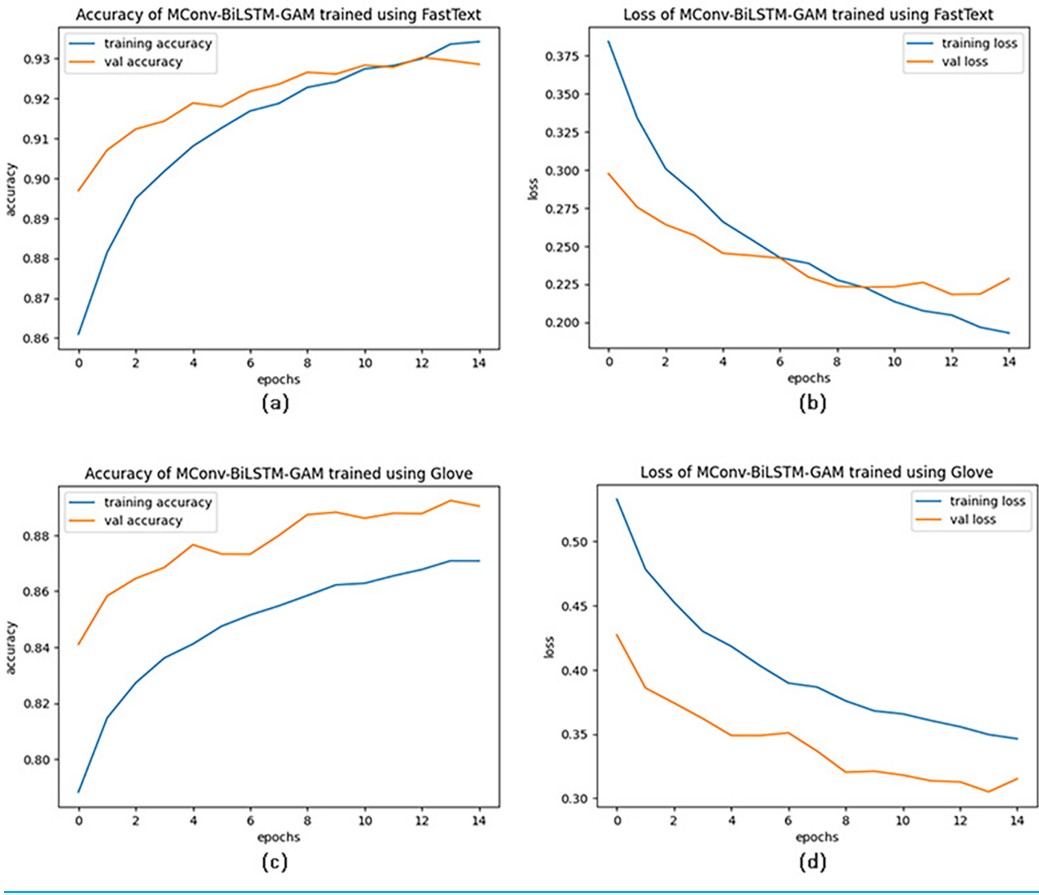

**Figure 4 Accuracy-Loss curves of MConv-BiLSTM-GAM model for FastText and GloVe word embedding techniques.**

regularization techniques and early stopping, which help improve the model's generalization performance.

Figure 5 presents the confusion matrices of the tested algorithms. The results indicate that the proposed model achieves the highest accuracy when used with FastText. By maintaining a more balanced and consistent performance across all sentiment classes, the proposed model minimizes misclassifications. In contrast, the CNN model demonstrates a higher error rate, particularly in the positive sentiment class, due to its limited ability to model sequential dependencies. LSTM and BiLSTM models outperform CNN by better capturing contextual relationships. However, models based on GloVe exhibit a higher misclassification rate compared to FastText. This discrepancy arises from GloVe's limited ability to represent semantic relationships between words. Overall, the proposed model, when combined with FastText, provides high accuracy and consistency, outperforming other models in sentiment classification tasks.

To compare training durations, the training times of different word embedding and classification models were examined over 15 epochs. The CNN model had the shortest training time with both embeddings, completing in 14 min using FastText and just 2 min

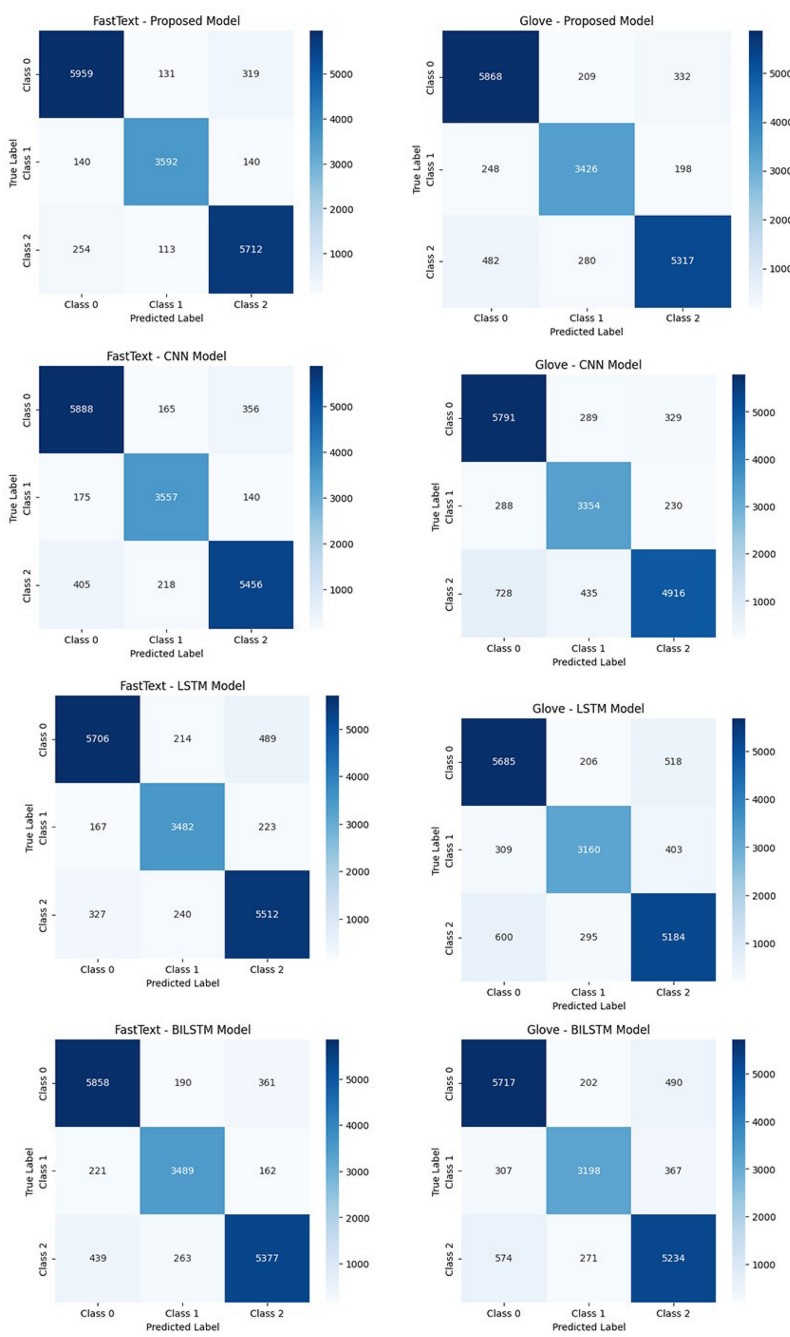

**Figure 5 The performance comparison of the classification models based on confusion matrices.**

with GloVe. In contrast, the MConv-BiLSTM-GAM model required the longest training time—249 min with FastText and 50 min with GloVe. The BiLSTM model also showed relatively longer durations, taking 166 min with FastText and 40 min with GloVe. These findings indicate that model complexity and the choice of word embedding method significantly affect training time.

**Table 2 The performance accuracy of alternative approaches for sentiment analysis on earthquakes.**

| Model | Accuracy (%) | Reported by |
|---|---|---|
| MConv-BiLSTM-GAM (FastText) | 93.32 | |
| MConv-BiLSTM-GAM (GloVe) | 89.26 | |
| MonkeyLearn | 0.63 | *Contreras et al. (2022)* |
| LR + TF | 0.88 | *Behl et al. (2021)* |
| MLP_W | 0.85 | *Behl et al. (2021)* |
| SVM | 81.2 | *Ruz, Henríquez & Mascareño (2020)* |
| Naïve Bayes | 74.2 | *Ruz, Henríquez & Mascareño (2020)* |
| MLP | 83.0 | *Behl et al. (2021)* |
| Random Forest-TF-IDF | 76.73 | *Yao & Wang (2020)* |
| Domain-adversial neural network (DANN) | 82.61 | *Yao & Wang (2020)* |
| SentiBERT-BiLSTM-CNN | 92.75 | *Song & Huang (2021)* |
| CNN + FastText | 86.0 | *Anthony, Hoi Ki Wong & Joyce (2024)* |
| BiLSTM + FastText | 84.0 | *Anthony, Hoi Ki Wong & Joyce (2024)* |
| LSTM + FastText | 81.0 | *Anthony, Hoi Ki Wong & Joyce (2024)* |
| Ensemble deep random vector functional links (edRVFL) | 84.29 | *Henríquez (2024)* |
| BERT-Feed forward neural network layers (BERT-FF) | 82.0 | *Alharm & Naim (2024)* |
| BERT-LSTM | 85.43 | *Alharm & Naim (2024)* |
| BERT | 71.0 | *Blomeier, Schmidt & Resch (2024)* |
| MNB + SMO | 87.9 | *Vo & Collier (2013)* |
| NB + SMOTE + GINI | 81.30 | *Gata et al. (2019)* |
| SVM + SMOTE + GINI | 81.03 | *Gata et al. (2019)* |
| SVM | 89.00 | *Saddam, Dewantara & Solichin (2023)* |
| XGBoost | 75.00 | *Detera et al. (2021)* |

In Table 2, the classification accuracy of the proposed approach was systematically compared with existing studies on SA of earthquake-related tweets in the literature. The results indicate that the proposed model demonstrated superior classification performance, achieving higher accuracy rates than other methodologies previously introduced in this research domain. This finding suggests that integrating the MConv-BiLSTM-GAM model effectively enhances sentiment classification accuracy.

Ablation studies were conducted to assess the individual contributions of MConv, BiLSTM, and GAM within the proposed MConv-BiLSTM-GAM architecture. The full model achieved the highest accuracy (93.32%) using FastText embeddings, confirming the strength of the integrated design. Removing GAM reduced accuracy to 92.33%, highlighting the importance of attention for semantic focus. Further removals of BiLSTM and MConv led to accuracies of 91.48% and 91.18%, respectively, indicating the value of sequential modeling and local feature extraction. Models using only LSTM or BiLSTM underperformed, emphasizing the necessity of combining convolutional, recurrent, and attention mechanisms. Overall, the results validate the effectiveness of the integrated architecture for sentiment analysis in noisy social media data during disasters.

## CONCLUSIONS

This study presents an effective deep learning-based sentiment classification approach by integrating a multi-convolution and bidirectional LSTM (MConv-BiLSTM) model with a GAM. FastText embeddings were used to vectorize tweets, enabling semantic-rich input representation. The architecture consists of three stages: MConv for extracting local features *via* convolutional layers with varying kernel sizes (2 and 3), BiLSTM for capturing temporal dependencies, and GAM to emphasize semantically significant information. Given the emotionally intense nature of disaster-related data, similar vocabulary may occur across sentiment classes. Therefore, the attention mechanism plays a critical role in distinguishing key emotional indicators. Experimental results confirm the model's high performance.

The sentiment analysis, based on a tri-polar classification (positive, negative, neutral), revealed that 37.08% of tweets were positive, 38.90% negative, and 24% neutral. Despite similar proportions, the underlying vocabulary differed notably: positive tweets emphasized support, solidarity, and reassurance, while negative tweets conveyed fear and destruction. Such emotional expression in disaster contexts can have lasting societal effects. The proposed model contributes to understanding collective emotional responses and provides actionable insights for decision-makers to support affected communities effectively.

Despite its promising outcomes, this study has several limitations. Relying solely on Twitter data may introduce bias, as platform users may not represent the broader affected population. Multilingual datasets also pose challenges due to language-specific sentiment expressions and syntactic differences, potentially affecting generalizability. Moreover, variations in user behavior and platform policies complicate cross-platform applicability. The model's exclusive focus on textual data limits its capacity to capture insights from multimodal content such as images and videos, which are commonly shared during disasters. Future research should consider integrating multimodal analysis, testing cross-platform adaptability, and enhancing linguistic and cultural versatility to improve the model's robustness.

### Funding

The authors received no funding for this work.

### Competing Interests

The authors declare that they have no competing interests.

### Author Contributions

- Serpil Aslan conceived and designed the experiments, performed the experiments, analyzed the data, performed the computation work, prepared figures and/or tables, authored or reviewed drafts of the article, and approved the final draft.

- Muhammed Yildirim conceived and designed the experiments, performed the experiments, analyzed the data, performed the computation work, prepared figures and/or tables, authored or reviewed drafts of the article, and approved the final draft.

## Data Availability

The code and dataset are available in the Supplemental File.

## Supplemental Information

Supplemental information for this article can be found online at http://dx.doi.org/10.7717/peerj-cs.2881#supplemental-information.

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
