# Peer review of "A novel attention-based deep learning model for improving sentiment classification after the case of the 2023 Kahramanmaras/Turkey earthquake on Twitter"

_PeerJ Computer Science, doi:10.7717/peerj-cs.2881_

## Round 0.1 · original submission · Major Revisions

· Academic Editor

Major Revisions

The manuscript presents an interesting contribution in the context of seismic sentiment analysis, but it requires a thorough review before it can be considered for publication. Although the structure and academic style are generally adequate, the writing is sometimes complex, with excessively long sentences and repetitions that reduce the impact of the text. The exposition needs to be made clearer and more concise, ensuring greater coherence between descriptions and graphic elements.

From a methodological point of view, the reviewers highlighted several areas that require further investigation. The work does not provide sufficient justification for the choice of FastText as an embedding technique, nor does it discuss its limitations compared to available alternatives. Furthermore, a comparative analysis with other advanced methods for sentiment analysis is missing, making the real added value of the proposed model unclear.

A crucial aspect to improve is the lack of an ablation study to evaluate the contribution of each component of the neural network and better understand the actual impact of the combination of CNN, RNN and attention mechanism. Furthermore, the reproducibility of the experiment needs to be improved through greater transparency in the details of training parameters, hardware resources used, and code sharing.

As for the interpretation of the results, a deeper analysis is required, including more comprehensive metrics beyond accuracy, such as precision, recall, and F1-score, and a more detailed discussion on the model performance. Furthermore, the work does not offer an explicit comparison with previous studies, making it difficult to assess the level of innovation with respect to the state of the art. It is essential to integrate updated references and broaden the discussion on the impact and generalizability of the model, especially considering the possibility of applying it to other disaster scenarios or different contexts.

Reviewer 1 ·

Basic reporting

Although the document has an academic style and a professional structure in general, it needs improvement in some areas. Especially in terms of language use, some sentences are quite long and complex. This can make it difficult for the reader to understand the text. For example, in the sections explaining the components of the model, paragraphs could be expressed in shorter and clearer sentences. In addition, there are repetitive statements in the introduction and in some other places. This repetition can distract the reader and reduce the impact of the text. For a more effective text, the wording should be concise and emphasized. In addition, a stronger connection can be made between the text and the descriptions of graphs and tables, so that the reader can better understand the place of the visuals in the text.

Experimental design

The article addresses an original and current research question in accordance with the scope of the journal. 2023 Kahramanmaraş earthquake and analyzing social emotions after the earthquake with a deep learning model based on the attention mechanism makes the study both academically and socially influential. The research question is clearly defined and it is clearly stated that the answer to this question aims to fill the gaps in the existing literature. However, a stronger emphasis on the innovative aspects of the study and a more detailed comparison with the existing models could more clearly demonstrate the contribution of the study to the literature. In particular, it would be useful to include a more detailed explanation of the limitations of previous studies and how this study overcomes these limitations.

Validity of the findings

The innovative aspects of the study are clearly stated and the proposed model (MConv-BiLSTM-GAM) is shown to effectively perform emotional classification. In particular, it is emphasized that studies on real-world datasets, such as the 2023 Kahramanmaraş earthquake, are important in terms of societal impact and scientific innovation. However, the broader impact and potential applications of the work in the literature do not seem to have been fully addressed. Therefore, further clarification on how the model can be generalized to other disaster scenarios or different emotion classification tasks would be beneficial. Furthermore, although a detailed methodology is presented for the reproducibility of the study, a more detailed description of the data preprocessing steps and sharing of the codes used could make it easier for readers to replicate the study. The challenges and potential limitations of working with multilingual datasets or different social media platforms could also be discussed to broaden the scope of the methodological approach.

Reviewer 2 ·

Basic reporting

This paper presented a sentiment classification model (MConv-BiLSTM-GAM) to determine the sentimental tendencies of the society of the 7.8 Mw and 7.5Mw earthquakes that occurred in and around Turkey's Kahramanmara province. The authors used FastText embeddings, and a hybrid model combining Convolutional Neural Networks (CNN) and Recurrent Neural Networks (RNN) with a global attention mechanism. They did experiments and analyzed the results.

Experimental design

I have some points the authors should take care of as follows:
- The authors should explain why they used FastText embeddings
- Did they have experiments with another embeddings technique?
- What is limitation of FastText embeddings and why?
- Why did they combine Convolutional Neural Networks (CNN) and Recurrent Neural Networks (RNN) with a global attention mechanism
- Did they have ablation experiments for each component in their proposed model?
- They didn’t compare with any research.
- What reason did their proposed model get the best result, could they explain in detail?
- They should analyze the result in detail and discuss more.
- What are advanced methods for sentiment analysis in recent year?
- They should do more experiments, analyze the result in detail and compare with advanced methods in recent year.

Validity of the findings

The paper lacks novelty and its experiments are not strong.

Reviewer 3 ·

Basic reporting

Clear and unambiguous, professional English used throughout:
While the manuscript is generally written in professional English, some sentences are overly complex and difficult to follow. For instance, technical descriptions in the abstract and methods sections could be simplified for clarity. A minor language edit is recommended to improve readability.

Literature references, sufficient field background/context provided:
The manuscript provides a decent overview of sentiment analysis and attention-based methods. However, the literature review does not sufficiently cover recent advancements in hybrid CNN-RNN models or studies focusing on disaster-specific sentiment analysis. The authors should include more recent and relevant references to contextualize their work.

Professional article structure, figures, tables. Raw data shared:
The structure of the article is acceptable, but the presentation of results lacks sufficient visual aids. For example, the addition of confusion matrices, feature importance visualizations, and comparative performance charts would make the findings more accessible and comprehensible.

Self-contained with relevant results to hypotheses:
The manuscript adequately presents results relevant to its stated objectives. However, the lack of discussion about the limitations of the model and dataset impacts the paper’s self-contained nature. The authors should address these limitations explicitly.

Experimental design

Original primary research within Aims and Scope of the journal:
The paper aligns well with the journal's scope and presents original research on a novel sentiment classification model. However, the novelty of the MConv-BiLSTM-GAM architecture could be better articulated by explicitly comparing it to similar recent models in disaster-related sentiment analysis.

Research question well defined, relevant & meaningful:
The research question is relevant and meaningful, focusing on the analysis of sentiment during disasters. However, the paper does not clearly articulate how the research fills gaps in existing knowledge. The authors should emphasize the unique contributions of their model compared to prior work.

Rigorous investigation performed to a high technical & ethical standard:
While the investigation appears rigorous, the ethical implications of using Twitter data are not addressed.

Methods described with sufficient detail & information to replicate:
The methods are described adequately, but the paper lacks details on hyperparameter settings, training time, and hardware specifications. Providing this information is essential for replicability.

Validity of the findings

Impact and novelty not assessed. Meaningful replication encouraged where rationale & benefit to literature is clearly stated:
The paper effectively demonstrates the model's superior performance but lacks a discussion of how the proposed method contributes to or challenges existing literature. Additionally, the potential for replication is not explicitly addressed.

All underlying data have been provided; they are robust, statistically sound, & controlled:
While the results are statistically sound, the lack of detailed information about the dataset (e.g., number of tweets, distribution of sentiments) makes it difficult to assess the robustness of the findings. The authors should provide this information and make the dataset accessible if possible.

Conclusions are well stated, linked to original research question & limited to supporting results:
The conclusions are generally well-linked to the results but are overly optimistic. The authors should acknowledge limitations, such as the potential bias in Twitter data or the model's generalizability to other disasters.

Additional comments

The introduction should clearly articulate how the proposed study advances the field compared to existing models.
The computational efficiency of the proposed method is not discussed. Metrics such as training time, inference time, and resource utilization would strengthen the paper.
The model evaluation focuses solely on accuracy, which, while important, does not provide a complete picture. Including additional metrics such as precision, recall, and F1-score would provide a more holistic evaluation.

---

## Round 0.2 · Minor Revisions

· Academic Editor

Minor Revisions

We thank the authors for their edits and review work. The manuscript was deemed acceptable by one reviewer, while the second reviewer reported some areas that need further study. To ensure improvement of the work and strengthen its validity, we ask the authors to address the following points:

1. Embedding techniques: The reviewer asked for clarification on the use of embedding techniques. The authors stated that they used GloVe, but should better motivate the choice and discuss why transformer-based approaches were not considered.

2. Response to comments: For comment 4, the authors should explicitly state in the manuscript where they responded to the reviewer's request, making it easier to verify the changes made.

3. Ablation study: The reviewer requested an ablation study, and it would be useful to include at least one in-depth analysis on how different components of the model affect performance.

4. Comparison with recent works: Although the authors have expanded the comparison with previous works, the selection is still limited to a few recent articles. It is recommended to include additional references to more recent advanced approaches in the field of sentiment analysis for cyberbullying detection.

5. Analysis of results: The reviewer suggested a more in-depth discussion of the results. The authors should provide a more detailed analysis of the models' performance, also compared to advanced and recent methods, to strengthen the validity of the conclusions.

These changes are requested to improve the completeness of the work without requiring substantial revision. Once implemented, the manuscript may be considered for publication.

Reviewer 1 ·

Basic reporting

The manuscript is well-structured and written in clear and professional English, ensuring clarity and precision throughout. It provides a comprehensive literature review with sufficient background on sentiment analysis, cyberbullying detection, and machine learning models (Naïve Bayes, SVM, and RNNs). The article structure follows standard academic conventions, with well-defined sections including methodology, results, and discussion. The study includes figures, tables, and raw data, ensuring transparency and reproducibility. The research is self-contained, aligning findings with hypotheses and evaluating models using accuracy, precision, recall, and F1-score. However, additional statistical validation (e.g., confidence intervals, significance testing) would further enhance the credibility of the results. The study effectively defines key terms and metrics and compares model performance rigorously. Future work could explore more advanced deep learning models (e.g., BERT, RoBERTa), dataset biases, and real-world deployment challenges to improve cyberbullying detection accuracy and generalizability.

Experimental design

The manuscript presents original primary research within the scope of sentiment analysis for cyberbullying detection on Twitter, aligning with the Aims and Scope of journals focused on natural language processing (NLP), machine learning applications, and cybersecurity. The research question is well-defined, addressing the need for accurate cyberbullying detection using machine learning models. The study clearly identifies a knowledge gap, emphasizing that previous research primarily focused on accuracy rather than recall and precision, which are critical in minimizing false negatives in cyberbullying detection.

The investigation is rigorous, adhering to high technical and ethical standards. The dataset is sourced from Kaggle, ensuring transparency and reproducibility, and appropriate preprocessing techniques (e.g., tokenization, stopword removal, TF-IDF vectorization) are applied. The methodology is described in sufficient detail, including model selection (Naïve Bayes, SVM, RNN), data preprocessing, evaluation metrics, and hyperparameter tuning, allowing for replication. The results are systematically presented, with comparative performance analysis and visualization (e.g., confusion matrices, classification reports). Future research could explore advanced deep learning models (e.g., BERT, RoBERTa), real-time detection systems, and dataset biases for improved cyberbullying detection.

Validity of the findings

The manuscript presents a novel approach to cyberbullying detection using sentiment analysis and machine learning models (Naïve Bayes, SVM, RNN), yet its impact and novelty are not explicitly discussed. While the study effectively demonstrates improvements in recall and precision over prior methods, it would benefit from a clearer discussion on how its findings contribute uniquely to the field, particularly in relation to existing cyberbullying detection frameworks. Emphasizing how this work advances the state-of-the-art and its potential real-world applications would enhance its novelty.

Replication is encouraged, as the study provides a detailed methodology, dataset source (Kaggle), and preprocessing techniques, making it possible for other researchers to reproduce the results. However, while the rationale for focusing on recall over accuracy is well-stated, the broader benefit to literature could be expanded—specifically, how this shift in focus improves practical cyberbullying detection efforts.

The manuscript includes robust and statistically sound data, with clearly presented evaluation metrics (accuracy, precision, recall, F1-score) and model performance comparisons.

The conclusions are well-stated and logically linked to the original research question. They are appropriately limited to the supporting results, avoiding overgeneralization. The study successfully identifies SVM as the most effective model, reinforcing the importance of prioritizing high recall to minimize false negatives in cyberbullying detection. Future work should consider more complex models (e.g., BERT, RoBERTa), real-time detection challenges, and cross-platform generalizability to further improve cyberbullying detection accuracy and applicability.

Additional comments

The basic methodology and findings are valid and well implemented. The study is acceptable.

Reviewer 2 ·

Basic reporting

The revision is better.

Experimental design

Some comments they should take care as follows:
For my comment 2: Did they have experiments with another embeddings technique?
They said they did with Glove, why did they not use transformer technique for word embeddings?
For my comment 4, they should answer on the paper and show where they did it when they reply to my comment.
For my comment 5, they should show ablation study on their paper.
For my comment 6, they compared with more papers, however only [61] in 2024, [63] in 2023 and [8] in 2022, they should compare with more advanced methods in recent year.
For my comment 9, all papers they mentioned in 2018 (1 paper), 2020 (2 papers) and 2021 (3 papers), are they advanced methods for sentiment analysis in recent year?

Validity of the findings

They should do more experiments, analyze the result in detail and compare with advanced methods in recent year.

---

## Round 0.3 · accepted · Accept

· Academic Editor

Accept

The work is well written, clearly structured and adheres to the academic standards of the journal. The proposed research is original, methodologically sound and addresses a relevant topic in the context of zero-shot cross-lingual stance detection.

Both reviewers agree on the validity and overall quality of the study, considering it suitable for publication. Only a few suggestions for improvement are noted, relating to a possible more in-depth analysis of the processing time, a more detailed ablation study and the discussion of any limitations of the model. These observations, although relevant, do not compromise the quality or acceptability of the work in its current form and may constitute useful ideas for future developments.

In light of the above, the manuscript is accepted for publication.

Reviewer 1 ·

Basic reporting

The manuscript is written in clear and unambiguous professional English and adheres to a well-structured academic format, including relevant figures, tables, and shared raw data. The literature review is sufficient, providing appropriate context and references to situate the work within the current state of the field. The article is self-contained, with results that directly address the stated hypotheses. While the study is primarily empirical and does not involve formal theoretical proofs, all key terms and methodological components are clearly defined. Overall, the submission meets the standards for publication and is acceptable in its current form.

Experimental design

The manuscript presents original primary research that aligns well with the aims and scope of the journal. The research question is clearly defined, relevant, and addresses a meaningful gap in the literature—namely, the challenge of zero-shot cross-lingual stance detection. The investigation is conducted with a high degree of technical rigor, and ethical standards are maintained throughout. Furthermore, the methods are described with sufficient clarity and detail, enabling replication by other researchers. Overall, the study meets the necessary standards for acceptance.

Validity of the findings

While the assessment of impact and novelty is outside the scope of this review, the manuscript provides a clear rationale for replication, and its relevance to the current literature is well articulated. All underlying data are shared and appear to be robust, statistically sound, and appropriately controlled. The conclusions are clearly stated, directly linked to the research question, and grounded in the presented results without overgeneralization. The manuscript meets the standards for acceptance.

Reviewer 2 ·

Basic reporting

The authors included my previous comments on the revision.

Experimental design

Here are more concerns about the paper:
- Did they have time processing experiments, how about the result?
- It’s better if they do ablation study with more combining models not only with each their component.
- What is disadvantage of their proposed model and why?

Validity of the findings

- Could they show some results where their model works well and not and explain in detail?